# Potential Capacity of *Candida wangnamkhiaoensis* to Produce Oleic Acid

**Alejandro Pérez-Rodríguez** [1]**, César Mateo Flores-Ortiz** [2,3]**, Griselda Ma. Chávez-Camarillo** [4]**,
Eliseo Cristiani-Urbina** [1,*] **and Liliana Morales-Barrera** [1,*]

[1] Departamento de Ingeniería Bioquímica, Escuela Nacional de Ciencias Biológicas, Instituto Politécnico Nacional, Av. Wilfrido Massieu s/n, Unidad Profesional Adolfo López Mateos, Ciudad de México 07738, Mexico; ibqaleprod@gmail.com

[2] Unidad de Biotecnología y Prototipos, Facultad de Estudios Superiores-Iztacala, Universidad Nacional Autónoma de México, Los Reyes Iztacala, Tlalnepantla, Estado de México 54090, Mexico; cmflores@unam.mx

[3] Laboratorio Nacional en Salud, Facultad de Estudios Superiores-Iztacala, Universidad Nacional Autónoma de México, Los Reyes Iztacala, Tlalnepantla, Estado de México 54090, Mexico

[4] Departamento de Microbiología, Instituto Politécnico Nacional, Escuela Nacional de Ciencias Biológicas, Prolongación de Carpio y Plan de Ayala s/n, Colonia Santo Tomás, Ciudad de México 11340, Mexico; gchcsepi@gmail.com

[*] Correspondence: ecristianiu@yahoo.com.mx (E.C.-U.); lmoralesb@ipn.mx (L.M.-B.);
Tel.: +52-55-5729-6000 (ext. 57835) (E.C.-U.); +52-55-5729-6000 (ext. 57827) (L.M.-B.)

**Abstract:** Oleic acid is increasingly required in many industries, causing the indiscriminate extension of land for the cultivation of certain agricultural products to extract their oil. The current contribution aimed to cultivate *Candida wangnamkhiaoensis* (CW) for the production of lipids and determine the profile of fatty acids in these lipids. The lipid yield was compared in the yeast when using glucose or glycerol as the substrate, in both cases being over 24%. The main fatty acids in the oil derived from CW were oleic, palmitic, stearic, and linoleic acid. The fatty acid composition of the oil from CW was very similar to that of avocado oil and resembled that of olive oil and palm oil. The advantages of cultivating CW include its relatively high percentage of oleic acid and the balance of other fatty acids, its capacity to generate lipids in a short time (48–72 h), the controlled environment of production (versus the variability of the cultivation of agricultural products), and the relatively limited surface area required. CW shows potential as an alternative and economical source of oleic acid for the food, drug, cosmetics, lubricant, and biofuel industries, and does not require the alteration of large extensions of land.

**Keywords:** *Candida wangnamkhiaoensis*; oleaginous yeast; lipid production; oleic acid; fatty acid profile; vegetable oil

## 1. Introduction

Lipids are either hydrophobic or amphipathic biomolecules that are characterized by being soluble in nonpolar solvents. They form parts of the cellular structures of organisms and participate in vital functions for maintaining cell activity [1].

Lipids contain fatty acids, which are classified as saturated (SFAs), monounsaturated (MUFAs), and polyunsaturated (PUFAs). Palmitic acid (C16:0), oleic acid (C18:1), and linoleic acid (C18:2) are the predominant SFA, MUFA, and PUFA, respectively, in vegetable oils [2]. MUFAs are fatty acids that are classified as nonessential in the diets of humans because they are synthesized by the organism [3]. However, their inclusion in the diet favors human health and may diminish the social cost of certain diseases [4].

Oleic acid is a MUFA that is known to be the principal component of several vegetable oils that have nutritional properties, especially avocado oil (42–51%) [5] and olive oil

(70–80%). Research on oleic acid has found antitumor effects [6], benefits to the cardiovascular system [7], that it participates in inducing increased sensitivity to insulin during the treatment of diabetes [8], and that it is involved in the regulation of the immune system [3].

The Mediterranean diet is considered to be healthy due to its content, mainly consisting of unprocessed cereals, legumes, fruits, vegetables, and olive oil [9]. Although olive oil is considered an essential part of this diet, the life cycle assessment of its production reveals that the cultivation of olives, the extraction of their oil, and the resulting waste lead to a significant adverse environmental impact [10,11]. Damage to ecosystems and a substantial water and carbon footprint have also been found in relation to the production of other vegetable oils that are important from a nutritional or industrial point of view, including palm [12], soybean, rapeseed, sunflower, coconut [13], and avocado oils [14].

In the search for alternatives to vegetable oils, microorganisms represent a promising source of lipids for use in the biofuel, pharmaceutical, and lubricant industries. Oleaginous microorganisms are capable of storing lipids as a source of carbon and/or energy, and such storage can constitute more than 20% of their cellular weight. Microbial lipids are denominated as single-cell oils (SCOs) [15].

One of the advantages of obtaining oils from microbial origins is that the life cycles of microorganisms are much shorter than those of plants. An additional advantage is the possibility of generating the oils *in vitro*, thus being without the influence of various factors involved in the elaboration of vegetable oils, including geography, seasonal harvests, and weather conditions [16]. Because the cultivation of microorganisms can be carried out with agricultural and industrial residues as substrates, it represents an economical alternative [15] that does not compete with the world demand for carbohydrates derived from food [17]. Furthermore, oils extracted from microorganisms could help to reduce the growing deforestation caused by the indiscriminate cultivation of palms and soybeans to obtain vegetable oils, which results in substantial harm to the ecosystems of tropical areas [15].

The yeast analyzed presently is *Candida wangnamkhiaoensis* (CW), which was initially identified as *Wickerhamia* sp. [18]. CW, which has scarcely been studied, belongs to the clade of *Hyphopichia* and is recognized for its capacity to assimilate glycerol [19]. In spite of a recent report, its metabolic and physiological capacities, as well as its potential in biotechnological processes, are unknown. Our group has investigated the capacity of CW to degrade starch and produce α-amylase, both systematically in batches [20] and continuously [21,22], and has described some of the biochemical and molecular properties of this enzyme [20]. During such research, it was observed that CW is capable of accumulating lipids, a property that has not, to our knowledge, been previously documented in the literature. Further evaluation of the potential of CW for generating lipids and fatty acids should certainly be of interest.

The aim of the current contribution was to examine the production of lipids by CW and determine the profile of fatty acids in these lipids. The substrates of the fermentation of CW were glucose and glycerol. The former is a conventional carbon source, and the latter is commonly found in the residues left by the production of biofuel [23], alcoholic beverages, and soaps [24].

Given that glycerol is a residue of some industrial processes [24], its use as a substrate would be a great benefit for the environment. Moreover, glycerol is considered to have a highly negative environmental impact because of its chemical oxygen demand (COD) of up to $1600 \text{ g O}_2 \text{ L}^{-1}$ [25].

The incubation of CW in culture medium with glucose or glycerol as the substrate generated an oil with a relatively large proportion of oleic acid. Regarding the composition of fatty acids, CW oil is very similar to avocado oil and resembles olive and palm oils. Because this microorganism has to date proven to be innocuous, it is a potential candidate for the development of an emerging technology, with a promising future in the production of single-cell oils as a raw material for the food, pharmaceutical, cosmetics, lubricant, and biofuel industries.

## 2. Materials and Methods

### 2.1. Microorganism

CW was isolated by the Industrial Microbiology Lab of the Escuela Nacional de Ciencias Biológicas (ENCB) at the Instituto Politécnico Nacional (Mexico City, Mexico) and provided to the authors. The strain was conserved in YPG medium with the following formulation: 10 g L$^{-1}$ yeast extract, 20 g L$^{-1}$ peptone from casein, 20 g L$^{-1}$ glucose, and 20 g L$^{-1}$ bacteriological agar, purchased from BD Bioxon (Mexico State, Mexico). The culture medium of CW for the production of lipids included the following salts: 0.1 g L$^{-1}$ KCl, 1.0 g L$^{-1}$ KH$_2$PO$_4$, 0.3 g L$^{-1}$ MgSO$_4$·7H$_2$O, 0.05 g L$^{-1}$ CaCl$_2$, 0.001 g L$^{-1}$ FeCl$_3$·6H$_2$O, and 1.0 g L$^{-1}$ (NH$_4$)$_2$SO$_4$ (JT Baker, Monterrey, Mexico). Glucose and glycerol at a concentration of about 10.0 g L$^{-1}$ (JT Baker, Monterrey, Mexico) were employed as a source of carbon and energy.

### 2.2. Preparation of the Inoculum

After putting an inoculation loop of CW into a beaker containing the culture medium along with either glucose or glycerol, incubation was carried out at 28 °C and under constant agitation at 110 rpm for 72 h. The resulting cellular suspension was centrifuged at 3500 rpm for 20 min to separate the biomass, which was washed three times with type II sterile water to eliminate the culture medium. The biomass pellet was resuspended in a little sterile water, and the concentration of biomass per mL of suspension was determined. A portion of each of the inoculates obtained with glucose or glycerol was used for optical imaging with confocal laser scanning microscopy (CLSM). The rest of the inoculate was subjected to a kinetic study of cell growth, substrate consumption, and lipid production.

### 2.3. Kinetics of Cell Growth, Substrate Consumption, and Lipid Production

The culture medium with the respective carbon source (glucose or glycerol) was placed in Erlenmeyer beakers. Subsequently, and based on the concentration of the biomass of the inoculum, a calculation was made of the quantity of CW to be added to the beakers to reach an initial biomass of 0.1 g L$^{-1}$. The yeast was then incubated under the aforementioned conditions for up to 120 h. Samples were taken every 24 h to monitor the concentration of residual substrate, biomass, and lipids. The fatty acid profile of the lipids was examined at the different sampling times.

### 2.4. Detection of Lipids by Confocal Laser Scanning Microscopy (CLSM)

A solution was prepared for optical imaging of the intracellular lipids of CW. It consisted of oil red O (1-([4-(xylylazo)xylyl]azo)-2-naphthol) in isopropyl alcohol at 98% (Sigma Aldrich, Toluca, Mexico State, Mexico). A sample of yeast growth (in glucose or glycerol) was smeared across a microscope slide. Oil red O was then added, and the slides were left to stand for 5 min. When dyed, the lipids of the cells generate fluorescence that can be detected using CLSM. The slides were inspected with a Zeiss LSM 710 NLO Multiphoton CLSM (Carl Zeiss Meditec AG, Jena, Germany) with a capacity in the range of 417–729 nm, utilizing Zeiss ZEN software and a Zeiss EC Plan-Neofluar 10×/0.3 objective lens.

### 2.5. Quantification of Biomass, Glucose, Glycerol, and Lipids

The yeast suspension was filtered through microfiberglass filters (Cytiva Whatman, St. Louis, MO, USA). The resulting biomass was washed twice with type II water, oven-dried at 60 °C for 24 h, and then quantified. The amount of residual glucose and glycerol in the filtrate was determined. While glucose was evaluated using the method of glucose oxidase and peroxidase [26], glycerol was assessed using the colorimetric method described by Bondioli and Della Bella [27]. Lipids were quantified in the biomass via gravimetry, with a solution of methanol–chloroform (2:1 *v/v*) as solvent [28].

### 2.6. Determination of the Profile of Fatty Acids and the Quantity of Each One

Once the oil was extracted [28], the fatty acid profile was established by the O'Fallon method [29], and then modified to esterify the fatty acids to methyl esters. This procedure was performed in the Biogeochemical Lab of the National Lab of Health in the Faculty of Graduate Studies Iztacala of the Universidad Nacional Autónoma de México, Mexico. The personnel operated an Agilent 6850 Series II Red System gas chromatography apparatus coupled with an Agilent Series 5975c mass spectrometer equipped with a flame ionization detector and an Agilent 19091S-433E capillary column of 30 m × 0.25 mm × 0.25 μm (Agilent Technologies, Inc., Santa Clara, CA, USA). The initial temperature (150 °C) was maintained for 2 min, followed by increments of 5 °C min$^{-1}$ until it reached 200 °C, and then of 3 °C min$^{-1}$ until it reached 260 °C. The injector and detector temperatures were fixed at 220 and 290 °C, respectively. Heptadecanoic acid served as the internal standard and helium served as the carrier gas at a flow of 1 mL min$^{-1}$.

### 2.7. Kinetic Parameters

With the data on the quantity of biomass, residual substrate, lipids, and fatty acids produced over time during the experiment, the kinetic parameters were calculated (Table 1).

**Table 1.** Kinetic parameters.

| Parameter | Formula | Nomenclature |
|---|---|---|
| Substrate consumption efficiency, $Ef$ (%) | $Ef = 100 \frac{S_0 - S_t}{S_0}$ | $S_t$: Residual concentration of the substrate at time $t$ (h) (grams of substrate L$^{-1}$) |
| Lipid yield based on biomass, $Y_{PX}$ (%) | $Y_{PX} = 100 \frac{P_t - P_0}{X_t - X_0}$ | $S_0$: Substrate concentration at the initial time $t = 0$ h (grams of substrate L$^{-1}$) |
| Lipid yield based on substrate, $Y_{PS}$ (g g$^{-1}$) | $Y_{PS} = \frac{P_t - P_0}{S_0 - S_t}$ | $P_t$: Lipid concentration at time $t$ (h) (grams of lipids L$^{-1}$) |
| Fatty acid yield based on biomass, $Y_{FAX}$ (mg g$^{-1}$) | $Y_{FAX} = \frac{FA_t - FA_0}{X_t - X_0}$ | $P_0$: Lipid concentration at the initial time $t = 0$ h (grams of lipids L$^{-1}$) |
| Volumetric biomass productivity, $R_x$ (mg L$^{-1}$ h$^{-1}$) | $R_X = 1000 \frac{X_t - X_0}{t - t_0}$ | $X_t$: Biomass concentration at time $t$ (h) (grams of biomass L$^{-1}$) |
| Volumetric lipid productivity, $R_P$ (mg L$^{-1}$ h$^{-1}$) | $R_P = 1000 \frac{P_t - P_0}{t - t_0}$ | $X_0$: Biomass concentration at the initial time $t = 0$ h (grams of biomass L$^{-1}$) $FA_t$: Fatty acid (FA) content at time $t$ (h) (milligrams of FAs L$^{-1}$) |
| Specific lipid productivity, $R_{px}$ (mg g$^{-1}$ h$^{-1}$) | $R_{PX} = 10 \frac{Y_{PX}}{t - t_0}$ | $FA_0$: Fatty acid (FA) content at the initial time $t = 0$ h (milligrams of FAs L$^{-1}$) |

### 2.8. Statistical Analysis

All determinations were made at least in duplicate for posterior statistical analysis with GraphPad Prism version 9.4.1 (GraphPad Software, San Diego, CA, USA). Significant differences between groups at each time point were evaluated by using two-way ANOVA with Tukey's multiple comparisons test, considering significance at $p < 0.05$.

## 3. Results and Discussion

### 3.1. Detection of Lipids by Confocal Laser Scanning Microscopy (CLSM)

The corresponding micrographs of CW grown with either glucose or glycerol are illustrated in Figures 1 and 2, respectively. The pseudomycelium that is characteristic of this species (described by Limtong [19]) can be observed with glucose (Figure 1a) but not with glycerol (Figure 2a) as the substrate.

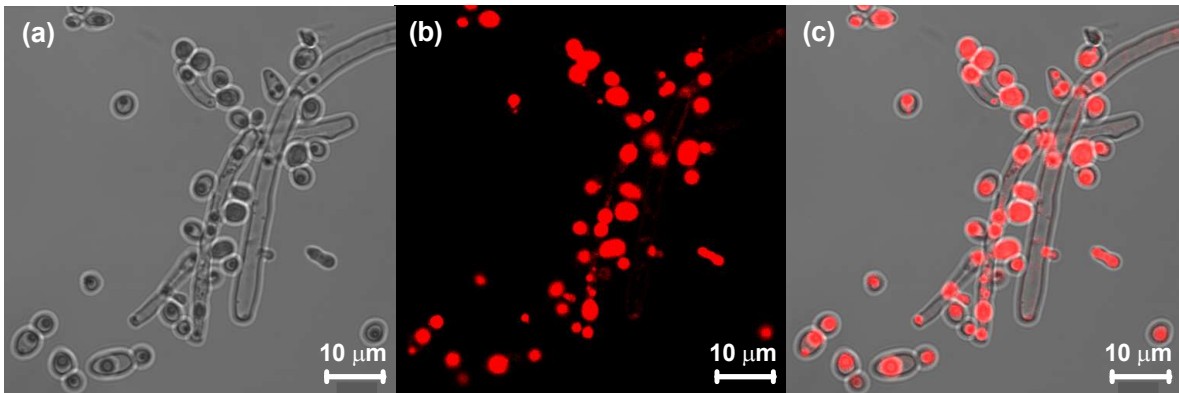

**Figure 1.** For *Candida wangnamkhiaoensis* cultivated in glucose, an optical micrograph (900×) (**a**), a confocal laser scanning microscope image (showing the fluorescence of the lipids) (**b**), and the overlap of the two images (**c**).

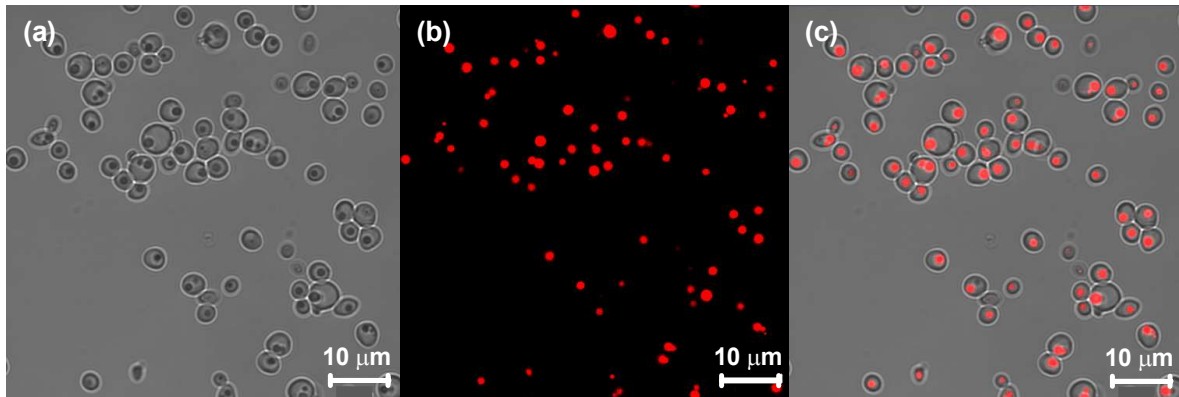

**Figure 2.** For *Candida wangnamkhiaoensis* cultivated in glycerol, an optical micrograph (900×) (**a**), a confocal laser scanning microscope image (showing the fluorescence of the lipids) (**b**), and the overlap of the two images (**c**).

There are yeasts that form pseudohyphae when cultivated with some substrates [30] and stress factors, the latter of which include a scarcity of the source of carbon or nitrogen, the presence of certain alcohols, osmotic shock, oxidative stress, and environmental extremes (e.g., in relation to pH and/or temperature) [31]. Although it is not known why CW presently exhibit pseudomycelium growth when cultivated in glucose and unicellular growth in glycerol, these results are in agreement with those reported for *Yarrowia lipolytica*, a yeast with biotechnological potential. The latter yeast also shows a predominantly mycelium form in a medium of glucose plus $(NH_4)_2SO_4$, and a yeast form in a medium of glycerol plus $(NH_4)_2SO_4$ [32].

By using oil red O staining and CLSM, intracellular lipids of CW were observed in pseudohyphae when cultivated with glucose (Figure 1b) and in ovoids when cultivated with glycerol (Figure 2b). The overlap of the images (Figures 1c and 2c) reveals a great quantity of lipids within the cellular structure of CW.

### 3.2. Kinetics of Cell Growth, Substrate Consumption, and Lipid Production by CW

The kinetics of cell growth, substrate consumption, and lipid production by CW are illustrated in Figure 3 for each of the two substrates, glucose (Figure 3a–c) and glycerol (Figure 3d–f).

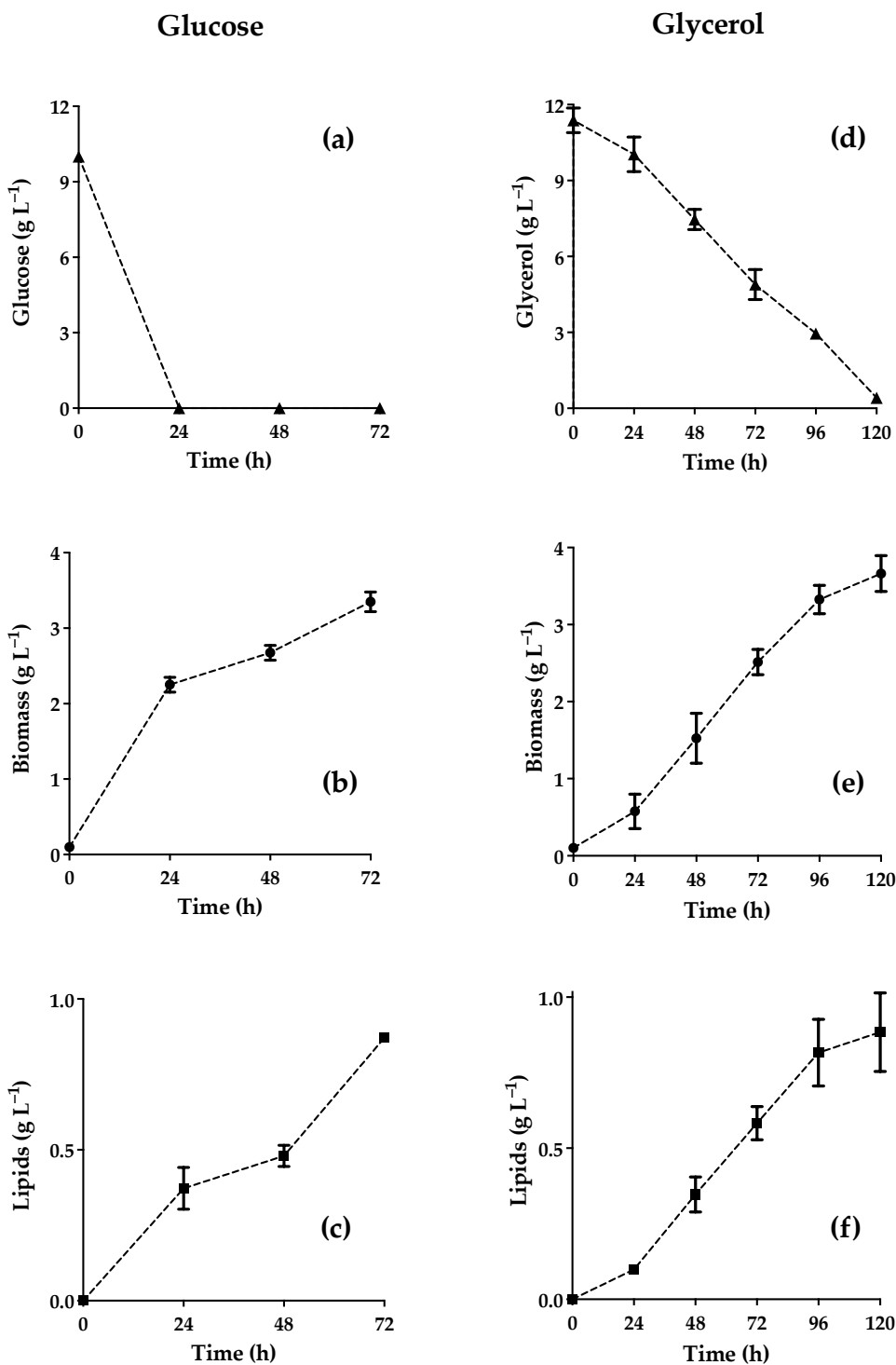

**Figure 3.** The concentration of substrate, biomass, and lipids over time during the incubation of *Candida wangnamkhiaoensis* with glucose (**a–c**) or glycerol (**d–f**) as the source of carbon.

Although the experiments began with similar initial concentrations of the substrate (10 g glucose $L^{-1}$ and 11 g glycerol $L^{-1}$) and were carried out for 72 h in each case, the time required for the total consumption of the substrate was distinct. Glucose was totally consumed by CW in 24 h (Figure 3a) (*Ef* = 100%), but glycerol had not yet been totally consumed at the end of 72 h. It was necessary to prolong the time of incubation to 120 h (Figure 3d) for the total consumption of glycerol to be attained (*Ef* = 96.35%). The glycerol molecule is better assimilated if amino acids, yeast extract, and peptone are added to the

culture medium to favor the growth of some yeasts [23,33]. However, none of these factors were added presently, which is probably the reason for the longer cultivation time required when using glycerol.

CW showed exponential growth during the first 24 h of cultivation with glucose (Figure 3b) and a slower growth rate afterwards when the substrate was nearly exhausted in the culture medium. Thus, CW was capable of growing and accumulating lipids under conditions of glucose scarcity, which may be due to its capability of intracellular accumulation of a carbohydrate reserve that can be used later [22]. That is, the yeast was previously prepared as a preinoculate grown in YPG, during which time it could accumulate a reserve of carbohydrates [20]. This is also a likely explanation for the lack of cell lysis after 24 h of growth in the glucose medium. The maximum concentration of biomass of CW cultivated in glucose was found at 72 h (3.35 g $L^{-1}$).

On the other hand, the growth of CW during the first 24 h in glycerol exhibited an adaptation period, with a posterior increase in the growth rate up to 96 h, followed by a decreased production of biomass. According to the analysis, using one-way ANOVA with Tukey's multiple comparisons test ($\alpha$ = 0.5), there was no significant difference between the concentrations of biomass in the samples of the yeast at 96 and 120 h in the culture medium with glycerol. Hence, the culture reached its stationary phase at 96 h, with its maximum concentration of biomass being approximately 3.6 g $L^{-1}$ (Figure 3e). This is nearly the same maximum quantity as that detected at 72 h when glucose was the substrate. No cellular lysis was found for CW cultivated in the glycerol substrate, which, as aforementioned, was also absent while cultivating the yeast with glucose.

Regarding lipids, their production increased during the entire time of the experiment. In culture medium with glucose (Figure 3c), the maximum concentration of lipids was observed at 72 h (0.871 ± 0.003 g $L^{-1}$). At this same time point, the concentration of lipids generated by CW in culture medium with glycerol (Figure 3f) was 33% lower (0.583 ± 0.05 g $L^{-1}$). Nevertheless, the maximum amount of lipids in CW cultivated with glycerol as the substrate (0.85 ± 0.12 g $L^{-1}$) was similar to that with glucose, although it was obtained between 96 and 120 h rather than at 72 h.

### 3.3. Kinetic Parameters of Cellular Growth, Substrate Consumption, and Lipid Production

The kinetic parameters were calculated for CW cultivated with glucose or glycerol as the substrate (Tables 2 and 3, respectively).

**Table 2.** Kinetic parameters of CW cultivated with glucose as the substrate.

| Time (h) | Ef (%) | X (g $L^{-1}$) | $Y_{PX}$ (%) | Lipids (g $L^{-1}$) | $Y_{PS}$ (g $g^{-1}$) | $R_X$ (mg $L^{-1}$ $h^{-1}$) | $R_P$ (mg $L^{-1}$ $h^{-1}$) | $R_{PX}$ (mg $g^{-1}$ $h^{-1}$) |
|---|---|---|---|---|---|---|---|---|
| 24 | 100 | 2.25 ± 0.05 [a] | 16.56 [a] | 0.37 ± 0.04 [a] | 0.03 ± 0.004 [a] | 93.75 ± 2.09 [a] | 15.52 ± 1.67 [a] | 6.90 ± 0.74 [a] |
| 48 | 100 | 2.68 ± 0.05 [b] | 18.26 [a] | 0.48 ± 0.02 [a] | 0.05 ± 0.002 [a] | 55.73 ± 0.99 [b] | 10.01 ± 0.42 [b] | 3.81 ± 0.16 [b] |
| 72 | 100 | 3.35 ± 0.07 [c] | 25.61 [b] | 0.87 ± 0.02 [b] | 0.09 ± 0.001 [b] | 46.53 ± 0.89 [c] | 12.09 ± 0.03 [ab] | 3.56 ± 0.01 [b] |

[a,b,c] For each parameter, there was no significant difference between the quantities marked with the same letter.

**Table 3.** Kinetic parameters of CW cultivated with glycerol as the substrate.

| Time (h) | Ef (%) | X (g $L^{-1}$) | $Y_{PX}$ (%) | Lipids (g $L^{-1}$) | $Y_{PS}$ (g $g^{-1}$) | $R_X$ (mg $L^{-1}$ $h^{-1}$) | $R_P$ (mg $L^{-1}$ $h^{-1}$) | $R_{PX}$ (mg $g^{-1}$ $h^{-1}$) |
|---|---|---|---|---|---|---|---|---|
| 24 | 11.79 | 0.58 ± 0.08 [a] | 15.35 [a] | 0.10 ± 0.01 [a] | 0.08 ± 0.02 [a] | 23.96 ± 3.31 [a] | 4.12 ± 0.21 [a] | 6.39 ± 0.45 [a] |
| 48 | 34.41 | 1.53 ± 0.11 [b] | 21.06 [b] | 0.35 ± 0.02 [b] | 0.09 ± 0.007 [a] | 31.77 ± 2.38 [b] | 7.22 ± 0.49 [b] | 4.38 ± 0.23 [b] |
| 72 | 56.97 | 2.51 ± 0.06 [c] | 23.21 [b] | 0.58 ± 0.02 [c] | 0.09 ± 0.004 [a] | 34.89 ± 0.81 [b] | 8.09 ± 0.31 [b] | 3.22 ± 0.13 [c] |
| 96 | 74.00 | 3.33 ± 0.06 [d] | 24.64 [b] | 0.82 ± 0.04 [d] | 0.10 ± 0.005 [a] | 34.64 ± 0.67 [b] | 8.86 ± 0.49 [b] | 2.67 ± 0.18 [cd] |
| 120 | 96.35 | 3.63 ± 0.08 [d] | 24.06 [b] | 0.88 ± 0.05 [d] | 0.10 ± 0.007 [a] | 30.52 ± 0.69 [ab] | 7.36 ± 0.44 [b] | 2.00 ± 0.14 [d] |

[a,b,c,d] For each parameter, there was no significant difference between the quantities marked with the same letter.

With glucose as the substrate, the maximum values of $Y_{PX}$ and $Y_{PS}$ were reached at 72 h of incubation (25.61% and 0.09 g $g^{-1}$, respectively) (Table 2). At this same time point, but with glycerol as the substrate, the values of these parameters were very similar

($Y_{PX}$ = 23.21% and $Y_{PS}$ = 0.09 g g$^{-1}$). With glycerol as the substrate, CW began the stationary phase of growth between 96 and 120 h. Maximum values of $Y_{PX}$ (24.64%) and $Y_{PS}$ (0.1 g g$^{-1}$) were observed at 96 h (Table 3), but there was no significant difference between the values of two consecutive 24 h measurements as of 48 h for $Y_{PX}$ and as of 24 h for $Y_{PS}$.

Based on the value of $Y_{PX}$ > 20%, CW can be considered an oleaginous microorganism [15]. Regarding lipid content as a percentage of biomass, the ~24% for CW is comparable to the 22% reported for *Rhodotorula* sp. LFMB 22 [34], inferior to the 43% found for *Yarrowia lipolytica* [35], superior to the 15.3% described for *C. oleophila* ATCC 20177, and superior to the 6.6% documented for *C. curvata* NRRL-Y 1511. All these studies utilized glycerol as the substrate [34].

The lipid content of oleaginous yeasts might be enhanced by modifying the composition of the corresponding culture medium. Several studies have sought to boost the amount of lipids produced per gram of biomass by modifying various factors. The factor proven to be particularly effective for this purpose is an increased carbon–nitrogen (C/N) ratio. For instance, Angerbauer et al. [36] improved the $Y_{PX}$ of *Lipomyces starkeyi* DSM 70295 from 40% to 68% by increasing the C/N ratio in the culture medium from 60 to 150. This implies a more limited nitrogen supply, which causes an oleaginous microorganism to produce more lipids. Theoretically, the production of 1 mol of triglyceride with glucose or glycerol as the substrate should afford a maximum yields of $Y_{PS}$ of 0.33 g g$^{-1}$ and 0.3 g g$^{-1}$, respectively [34]. In practice, the yield is expected to be less than the theoretical value.

Low C/N ratios were used in the current investigation, being 18.86 g g$^{-1}$ (22 mol mol$^{-1}$) for glucose and 20.2 g g$^{-1}$ (23.6 mol mol$^{-1}$) for glycerol. Increasing the C/N ratio in the culture medium of CW could possibly boost the yields of $Y_{PX}$ and $Y_{PS}$, which were approximately 24% and 0.1 g g$^{-1}$, respectively, for both substrates. On the other hand, Yang et al. [37] demonstrated that the yield of lipids ($Y_{PS}$) produced by *Rhodosporidium toruloides* Y4 rose from 0.16 g g$^{-1}$ to 0.21 g g$^{-1}$ after boosting the concentration of glycerol in the culture medium from 20 to 150 g L$^{-1}$.

When comparing CW cultivated with glucose or glycerol as the substrate, the yields of $Y_{PX}$ and $Y_{PS}$ did not show any significant differences ($p$ > 0.05) at 72 h of incubation. Considering the kinetic profile of the yeast, the cultivation time does indeed affect growth and the quantity of lipids obtained, as can be appreciated by the values of productivity (Tables 2 and 3). With glucose rather than glycerol as the substrate, CW exhibited greater productivity of biomass ($R_X$) and lipids ($R_P$) at 24 h of incubation ($R_X$ = 93.75 mg L$^{-1}$ h$^{-1}$ versus 23.96 mg L$^{-1}$ h$^{-1}$ and $R_P$ = 15.52 mg L$^{-1}$ h$^{-1}$ versus 4.12 mg L$^{-1}$ h$^{-1}$, respectively). At the same time point, the specific productivity of biomass ($R_{PX}$) was not significantly different between these two substrates, with values close to 6 mg g$^{-1}$ h$^{-1}$. The value of $R_{PX}$ decreased over time for CW cultivated in either substrate. For CW incubated with glucose, the productivity values of $R_X$ and $R_P$ declined over time. For CW incubated with glycerol, in contrast, the maximum values of $R_X$ and $R_P$ were reached between 72 and 96 h ($R_X$ = 34 mg L$^{-1}$ h$^{-1}$, $R_P$ = 8 mg L$^{-1}$ h$^{-1}$), although the difference between each consecutive 24 h measurement was not significant as of 48 h. At all measurement times, the productivity indexes of $R_X$, $R_P$, and $R_{PX}$ were lower for CW cultivated with glycerol versus glucose as the substrate. A similar result was reported for the growth of *Meyerozyma guilliermondii* BI281 in glucose or glycerol [38].

### 3.4. Fatty Acid Profile

The fatty acid profile of an oleaginous microorganism varies in accordance with diverse factors, such as the variety of the microorganism, the carbon source utilized, and the time of incubation. The determination of this parameter is important because the fatty acid content is an indicator of the potential use of the oil obtained [39,40].

The principal fatty acids detected in the oil derived from CW after cultivation in either glucose or glycerol as the substrate were oleic (18:1), palmitic (16:0), stearic (18:0), linoleic (18:2), and palmitoleic acids (16:1). These fatty acids have also been found in the lipid profiles of other oleaginous yeasts, including *Rhodosporidium toruloides* Y4 [37] and

*Yarrowia lipolytica* [41] cultivated with glycerol as the substrate, *Rhodotorula kratochvilovae* SY89 grown on molasses [42], *Yarrowia lipolytica* CBS 6303 incubated with glucose [43], and *Cryptococcus laurentii* 11 with whey and molasses as carbon sources [44].

Based on the concentrations of fatty acids obtained, a calculation was made of the yields of the different fatty acids in relation to biomass ($Y_{FAX}$) (Figure 4), finding higher values with glucose (Figure 4a) versus glycerol (Figure 4b) as the substrate. At 48 h of incubation, for example, the production of oleic acid by CW was 76.35% greater with glucose ($Y_{FAX}$ = 52.94 mg g$^{-1}$) than glycerol ($Y_{FAX}$ = 30.02 mg g$^{-1}$).

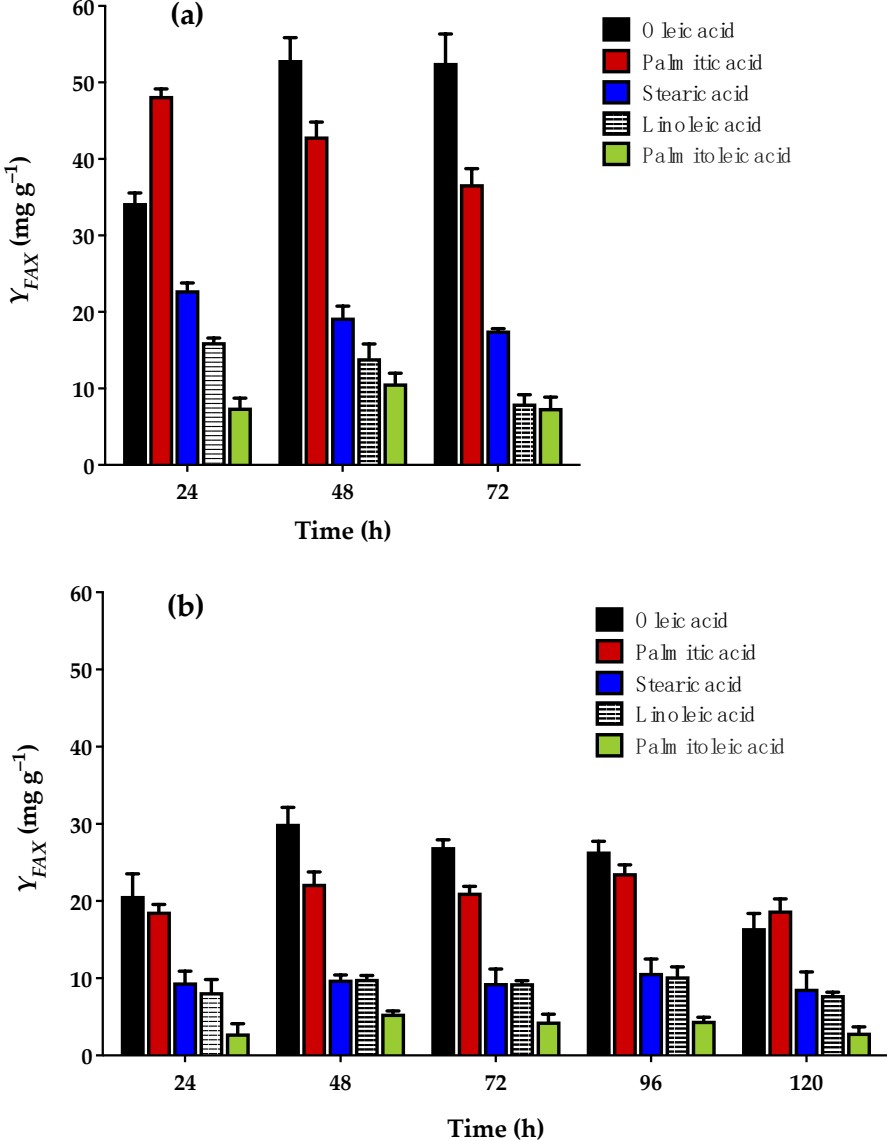

**Figure 4.** Yield of fatty acids per gram of biomass of *Candida wangnamkhiaoensis* ($Y_{FAX}$), generated with glucose (**a**) or glycerol (**b**) as the substrate.

At almost all times assayed with each substrate, oleic acid displayed the highest yield of any fatty acid, followed by palmitic acid. Similar results have been reported for the lipid profile of other yeasts cultivated in glucose or glycerol (Table 4).

**Table 4.** Composition of the main fatty acids (expressed as a percentage of total lipids) produced by different oleaginous yeasts with glucose or glycerol as the carbon source.

| Yeast | Carbon Source | C/N Ratio (mol mol$^{-1}$) | Initial pH | Temp (°C) | $Y_{PX}$ (%) | Incubation Time (h) | Fatty Acid Composition (% *w/w*) | | | | | | | Ref. |
|---|---|---|---|---|---|---|---|---|---|---|---|---|---|---|
| | | | | | | | Myristic (14:0) | Palmitic (16:0) | Palmitoleic (16:1) | Stearic (18:0) | Oleic (18:1) | Linoleic (18:2) | Linolenic (18:3) | |
| *Candida inconspicua* IGII | YPGlc | NR | 5.6 | 28 | 32 | 96 | ND | 16.61 | 2.18 | 12.28 | 48.5 | 18.28 | 2.15 | [45] |
| *Candida wangnamkhiaoensis* | Glucose | 23.7 | 6.0 | 30 | 25 | 72 | ND | 30 | 6 | 14 | 43 | 7 | ND | This work |
| *Cryptococcus aerius* UIMC65 | Glucose | 40 | 5.5 | 28 | 77 | 72 | 0.2 | 1.7 | 21.21 | 7.67 | 61.39 | 6.17 | NR | [40] |
| *Debaryomyces hansenii* 1 | YPGlc | NR | 5.6 | 28 | 46 | 96 | ND | 15.94 | 3.8 | 11.5 | 46.2 | 15.21 | 2.53 | [5] |
| *Lipomyces starkeyi* AS 2.1560 | Glucose | 1540 * | 6.0 | 30 | 59.3 | 120 | 0.2 | 35.6 | 3.8 | 6.0 | 53.1 | 0.7 | 0.4 | [46] |
| *Yarrowia lipolytica* JMY 794 | Glucose | 200 | 6.2 | 28 | 9 | 100 | ND | 19.3 | 11.3 | 10.2 | 41.1 | 18.1 | ND | [47] |
| *Candida oleophila* ATCC 20177 | Pure glycerol | 66 | 6 | 28 | 15.3 | 150 | NR | 12.9 | 2.5 | 6.6 | 65.6 | 11.0 | NR | [34] |
| *Candida wangnamkhiaoensis* | Pure glycerol | 23.7 | 6.0 | 30 | 24 | 48 | ND | 29 | 7 | 13 | 39 | 13 | ND | This work |
| *Meyerozyma guilliermondii* BI281A | Pure glycerol | 407 | NR | 28 | 34.97 | 120 | NR | 25.75 | ND | 34.17 | 48.76 | NR | 4.2 | [38] |
| *Pichia kudriavzevii* MTCC 5493 | Crude glycerol | NR | 5.5 | 28 | 18.6 | 110 | NR | 29.4 | NR | 8.9 | 41.9 | 9.2 | NR | [48] |
| *Rhodosporidium toruloides* Y4 | Pure glycerol | NR | 5.5 | 30 | 34.8 | 120 | 1.4 | 27.8 | 0.6 | 21.8 | 43.8 | 2.9 | 1.2 | [37] |
| *Rhodotorula* sp. LFMB22 | Pure glycerol | 66 | 6 | 28 | 22 | 187 | NR | 21.7 | 1.1 | 7.4 | 55.9 | 12.4 | NR | [34] |

NR, not reported; ND, not detected; YPGlc, yeast–peptone medium supplemented with glucose. * Value reported in a culture medium of 70 g glucose L$^{-1}$ and 0.1 g (NH$_4$)$_2$SO$_4$ L$^{-1}$.

The production of fatty acids allows oleaginous microorganisms to confront environmental factors capable of causing stress. Whereas long-chain fatty acids generally afford the membrane with rigidity and stability, short-chain fatty acids usually give greater fluidity [49]. Saturated fatty acids can have an important role in the polarity of the membrane [50]. They also favor membrane resistance to solvents and extreme temperatures due to their relatively high level of van der Waals interactions. Meanwhile, unsaturated fatty acids increase the fluidity of the membrane [51]. Oleic acid produces yeast that is more resistant to cold [52]. Palmitoleic and oleic acids, synthesized from palmitic and stearic acids, confer yeasts with tolerance to ethanol and high temperatures [53–55].

The fatty acids identified in CW may be instrumental in diverse industries for the production of food, biofuel, cosmetics, drugs, polymers, lubricants, and dispersants [56]. Moreover, in the area of food, proposals have been made for the creation of oleogels [57] as a substitute for fats such as cocoa butter [58], and for the use of certain combinations of fatty acids to provide an additional source of energy to cattle [59].

### 3.5. Comparison of the Composition of the Fatty Acids between CW and Some Vegetable Oils

Because the main fatty acids derived from yeasts are oleic, palmitic, and stearic acids, many authors have emphasized the similarity of this profile with that of vegetable oils, especially those utilized for the production of biodiesel and oleochemicals [40,45,60–62]. Radar charts were constructed presently to visualize and compare the composition of fatty acids between the oil currently obtained from CW and some vegetable oils that are rich in oleic acid (Figure 5), including avocado oil (from *Persea americana* Mill) [63] (Figure 5a), olive oil (Figure 5b), and palm oil (Figure 5e,f). Peanut oil is a common raw material in the food industry (Figure 5c), whereas oil from soybeans (Figure 5d), palm fruit (Figure 5e), and palm kernels (Figure 5f) have many applications in industry [64]. For the comparison of lipid profiles, the data on vegetable oils were found in previous reports [60,61], and the data on the oil derived from CW were taken from the present study, being the percentage of fatty acids in the yeast cultured with glucose or glycerol as the substrate at incubation times of 72 and 48 h, respectively. At these times, the concentration of oleic acid and the yield of $Y_{PX}$ (>20%) reached their maximums. At 48 h, as aforementioned, there were no significant differences between CW cultivated with glucose or glycerol in relation to yields of fatty acids ($Y_{PX}$ and $Y_{PS}$) or productivity ($R_x$ and $R_P$).

The composition of fatty acids of CW (especially when cultivated in glycerol) is very similar to that of avocado fruit (Figure 5a) and resembles that of olives (Figure 5b) and palm fruit (Figure 5e). Avocado, olive, and palm fruit oil contain high percentages of MUFAs, most notably of oleic acid. Therefore, they share many properties capable of diminishing the rate of progression of some degenerative and/or chronic diseases [65]. On the other hand, peanuts (Figure 5c) and soybeans, being legumes (Figure 5d), have an abundance of linoleic acid, in contrast to the greater amount of palmitic and stearic acids in CW.

From the oleaginous palm *Elaeis guineensis*, oil is extracted from the seed (palm kernel) and the mesocarp of the fruit (palm fruit). Oil from the palm fruit, known as crude palm oil or red palm oil, has properties that lend themselves to more numerous applications [66]. Its content of palmitic acid is greater than that obtained from CW (Figure 5e). The lipid profile of the oil in CW is more similar to that of palm fruit oil than palm kernel oil due to the greater presence of lauric and myristic acids in the latter (Figure 5f).

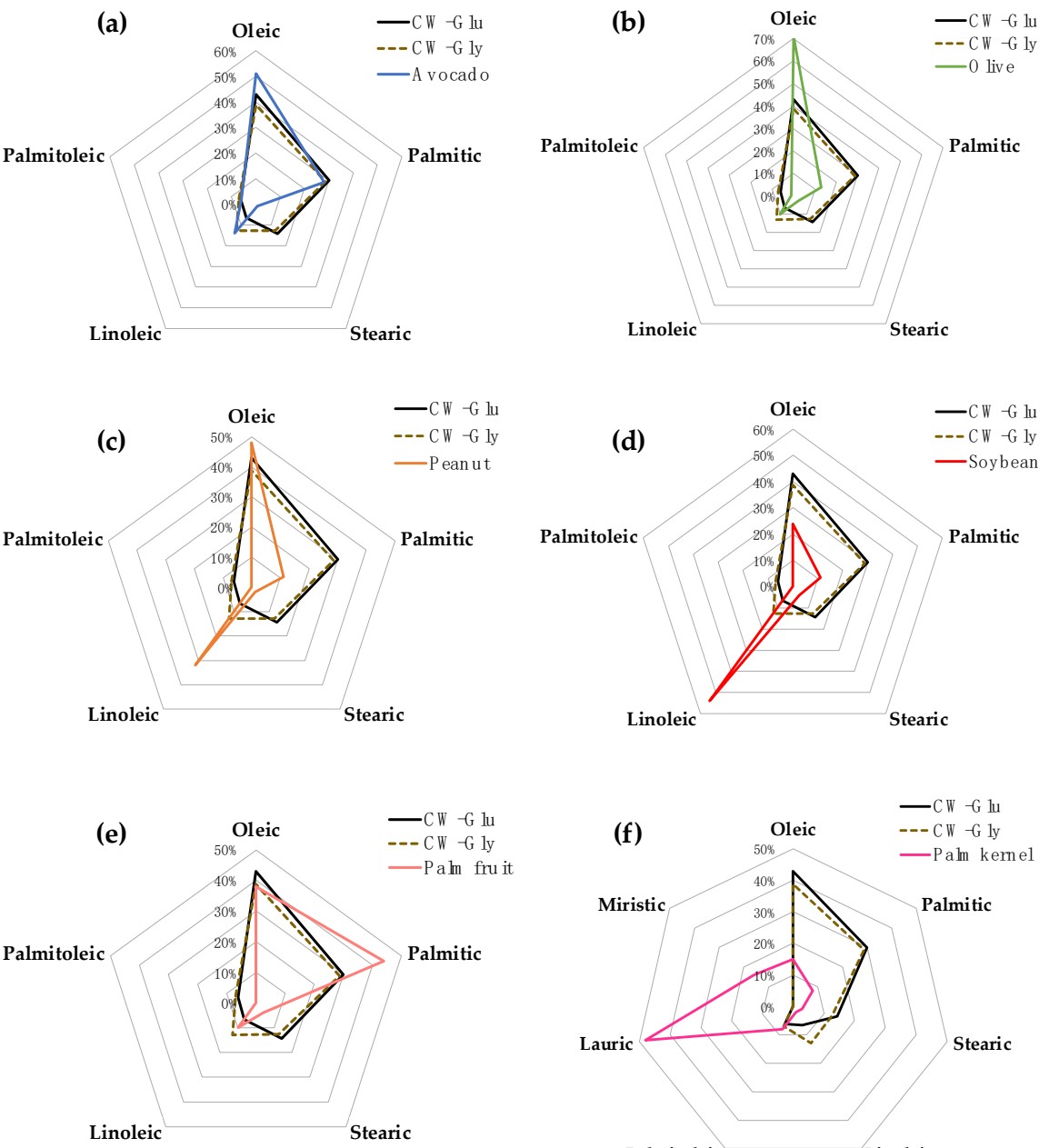

**Figure 5.** Regarding the composition of fatty acids (% *w/w*), comparisons were made between CW cultivated with glucose (CW-Glu) for 72 h or glycerol (CW-Gly) for 48 h and different vegetable oils extracted from avocado fruit (**a**), olives (**b**), peanuts (**c**), soybeans (**d**), palm fruit (**e**), and palm kernels (**f**).

*3.6. Plausible Uses of the Oil in CW Based on Its Similarity to Some Vegetable Oils*

According to the previous analysis of composition, the vegetable oils most similar to the oil derived from CW are firstly avocado oil and secondly olive oil and palm fruit oil. Hence, the possible applications of the oil contained in CW will be discussed within the context of the uses of these vegetable oils.

Only 3% of the production of avocados around the world is designated for the preparation of oil, which is considered a high-cost gourmet product [67]. This oil would have good potential as a raw material for widespread use in various industries because of its balance of oleic, palmitic, and stearic acids [65]. The only drawback is its limited production. Owing to their health benefits, both avocado and olive oils are mainly consumed in salads,

dressings, and other foods. However, they are also utilized in the elaboration of cosmetics, pharmaceutical products, and skin care products [65,68]. In many of these applications, the oil obtained from CW could probably serve as an alternative. Whereas 97% of avocados harvested are destined for sale as fresh fruit [66], over 90% of olives and palm fruit are processed for the extraction of their oil [69].

The worldwide production of palm oil during 2020 was estimated to be 70 million tons [70], of which 90% was destined for the preparation of margarine, spreads, confectionery fats, ice cream, emulsifiers, and ghee. Given the versatility provided by its balanced content of saturated (e.g., palmitic acid) and unsaturated fatty acids (e.g., oleic acid), it is a common raw material in the fabrication of cosmetics, tooth paste, and biofuel [66,71].

Considering the similar composition between the oil of CW and palm fruit oil, the former could potentially serve as a substitute, thus mitigating the negative effects caused by the massive cultivation of palms in tropical forests. The resulting deforestation and the improper disposal of waste material threaten biodiversity and create environmental damage [72,73].

Taking into account the similarity of the lipid profile of the oil from CW, avocado oil, olive oil, and palm oil, an analysis was made of the characteristics of the crops responsible for yielding the vegetable oils (Table 5), including the yield and productive age of the corresponding trees, and the overall worldwide production of each fruit.

**Table 5.** Characteristics of avocado, olive, and palm oil crops.

| Crop | World Crop Production (Million Metric tons year$^{-1}$) | Crop Yield (tons ha$^{-1}$ year$^{-1}$) | Pulp Content (%) | Oil Contained in the Pulp (%) | World Oil Production (Million Metric tons year$^{-1}$) | Oil Yield (tons ha$^{-1}$ year$^{-1}$) | Productive Time of Trees (Years) | |
| --- | --- | --- | --- | --- | --- | --- | --- | --- |
| | | | | | | | To Begin Yield | Maximum Productive Age |
| Avocado (*Persea americana* Mill.) | 8.06 [74] | 9.6 [67] | 60 [5] | 7–37 [75] | 0.371 [65] | NDA | 3–4 [76] | 30–50 [77] |
| Olive (*Olea europaea* L.) | 21.33 [78] | 0.5–12 [68] | 65–85 [79] | 30–40 [80] | 3.1 [81] | 9.0 [82] | 5–6 [68] | >100 [68] |
| Oil palm (*Elaeis guineensis* Jacq.) | 418.4 [83] | 14.56 [83] | 35–75 [84] | 45–70 [66,71,84] | 73.8 [85] | 1.92–15.49 [70,85,86] | 3–6 [87] | 25 [87] |

NDA, no data available.

Of the three crops examined herein, the one with the most abundant worldwide production is palm oil, with olive oil in second place and avocado oil in last place. Palm fruit has a great quantity of pulp that contains a large amount of oil. However, the yield of oil is variable from year to year and is contingent on many factors. The trees take 3–6 years to mature (to be able to give fruit) and have a productive life of 25 years (Table 5). The yield per hectare depends on the age of the trees and the diseases affecting them, as well as the maturity of the fruit, the harvest season, the characteristics of the soil and climate, the hydrological conditions, and the methods of oil extraction, among other factors [65,68,70,75,86,87].

Due to the variability of the yields of oil from the crops shown in Table 5, it is difficult to formulate a comparison with the oil obtained from CW. To address this problem, average values were used to construct the graph in Figure 6. In the case of avocados, the calculation was made with the data in Table 5 and the fact that 3% of production is destined to the extraction of oil. To determine the yield per hectare of CW, the diameter of the horizontal surface of the cultivation beaker was taken as the surface area, and the production time considered was 72 h with glucose and 48 h with glycerol as the substrate.

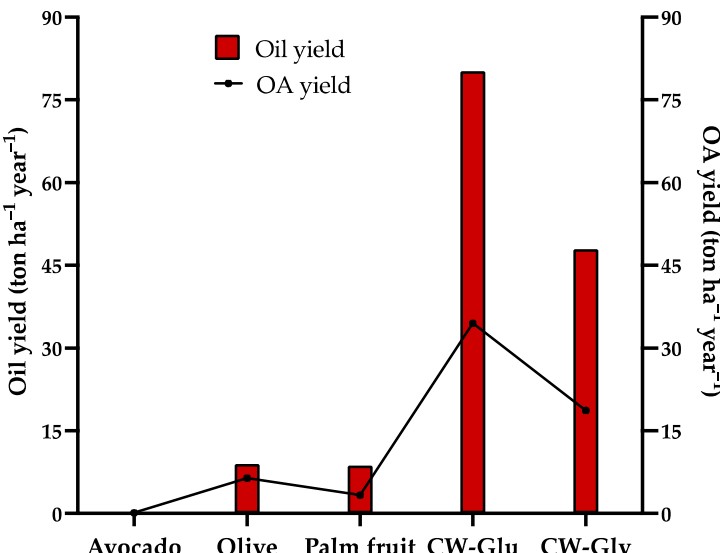

**Figure 6.** Comparison of the yields of oil and oleic acid (OA) obtained from avocado fruit, olives, and palm fruit, as well as *Candida wangnamkhiaoensis* cultured in glucose (CW-Glu) and glycerol (CW-Gly).

Given the importance of oleic acid in diverse industries, its yield was graphed (Figure 6) for the different crops and the present yeast cultures. The proportion of oleic acid in each type of oil was taken into account, constituting 51% of the total weight of avocado oil [63], 71% of olive oil [64], and 38% of palm oil [64]. The percentage of oleic acid in the composition of the oil derived from CW cultivated in glucose or glycerol was based on this study (43% and 39%, respectively). As can be appreciated (Figure 6), the potential yields per hectare are far superior for oil and oleic acid obtained from CW versus avocados, olives, and palm fruit.

According to the current calculations, CW cultivated in glucose yields an 8-fold greater amount of oil than that afforded by palm fruit or olives, and a 4.4-fold greater quantity of oil than that furnished by CW cultivated in glycerol. Although glycerol leads to a lower yield of lipids, it is a byproduct of some industries, thus conferring an economic benefit to its use. With either substrate, CW generates an oil with a relatively high proportion of oleic acid in only 48–72 h, and the yields are constant, being provided in a controlled environment (unlike the variability of the yields of avocados, olives, and palm fruit).

Considering the similarity of the lipid profile of oil derived from CW to that of certain vegetable oils, the lipids obtained from this microorganism could possibly be utilized in many industries as an economical alternative. The elaboration of oil from CW would contribute to the avoidance of shortages of vegetable oils and key fatty acids (e.g., oleic acid). It would also eliminate the need for the indiscriminate extension of land dedicated to the cultivation of products converted into vegetable oil, which would therefore reduce the displacement of endemic flora and the resulting alteration of ecosystems around the world.

## 4. Conclusions

When cultivating CW with glucose or glycerol as the substrate under the conditions of the present study, the microorganism was able to grow quickly and generate a lipid content above 21% of its biomass, thus becoming an oleaginous yeast. Because the corresponding lipids are rich in oleic acid, the cultivation of CW may serve as an alternative source of this fatty acid for the food, pharmaceutical, cosmetics, lubricant, and biofuel industries. To our knowledge, the capacity of CW to produce fatty acids has not been previously reported. The composition of fatty acids in the oils obtained from CW is very similar to that of avocado oil (e.g., stearic, palmitic, and oleic acids). Moreover, the oil of CW resembles that extracted from olives and palm fruit. These three vegetable oils are important in the production of food, drugs, cosmetics, and oleochemicals, and, in some cases, biofuel. The annual yield

of lipids (particularly of oleic acid) derived from CW per hectare could be up to 8.2-fold greater than the yields provided by the cultivation of avocado, olive, or palm trees. Because the oil from CW can be produced at a low cost, it represents a plausible alternative source for the lipids required in a great variety of industries, and, at the same time, could mitigate the environmental damage caused by constantly expanding cultivations. Future research is needed to improve the yield of lipids in CW by modifying the conditions of cultivation. For example, the C/N ratio in the culture medium could be increased and alternative reaction systems (distinct from those utilized presently) could be explored.

**Author Contributions:** Conceptualization, E.C.-U. and L.M.-B.; methodology, E.C.-U., C.M.F.-O. and L.M.-B.; software, A.P.-R. and L.M.-B.; validation, L.M.-B. and E.C.-U.; formal analysis, A.P.-R., E.C.-U. and, L.M.-B.; investigation, A.P.-R.; resources, L.M.-B. and E.C.-U.; writing—original draft preparation, L.M.-B. and E.C.-U.; writing—review and editing, L.M.-B. and E.C.-U.; visualization, L.M.-B.; supervision, E.C.-U., C.M.F.-O., G.M.C.-C. and L.M.-B.; project administration, L.M.-B. and E.C.-U.; funding acquisition, L.M.-B. and E.C.-U. All authors have read and agreed to the published version of the manuscript.

**Funding:** This research was funded by the Instituto Politécnico Nacional, Secretaría de Investigación y Posgrado (grant numbers: SIP 20230270 and SIP 20231441).

**Institutional Review Board Statement:** Not applicable.

**Informed Consent Statement:** Not applicable.

**Data Availability Statement:** All relevant data are within the paper.

**Acknowledgments:** The authors acknowledge the technical support provided by the Centro de Nanociencias y Micro y Nanotecnologías at the Instituto Politécnico Nacional. The CONACyT (Consejo Nacional de Ciencia y Tecnología) awarded a graduate scholarship to the coauthor A.P.-R., G.M.C.-C., E.C.-U. and L.M.-B. received research grants from the EDI-IPN, COFAA-IPN, and SNI-CONACYT (National System of Researchers). C.M.F.-O. is a member of the SNI-CONACyT. We thank Bruce Allan Larsen for proofreading the manuscript.

**Conflicts of Interest:** The authors declare that they have no conflict of interest.

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
