# Peer review of "Potential Capacity of Candida wangnamkhiaoensis to Produce Oleic Acid"

_fermentation, doi:10.3390/fermentation9050443_

Round 1

Reviewer 1 Report

This study reports the cultivation of Candida wangnamkhiaoensis (CW) to produce lipids from glucose and glycerol. The lipids are characterized regarding the fatty acid profiles obtained and compared with other traditional lipid sources. The methodology is well described and executed, and the results are clearly and carefully exposed. The manuscript is well structured and is easy to follow and understand.

Nevertheless there is no further exploration of factors that might contribute to improve the lipids production (which I assume was not the purpose of the study) – example, the increase of the carbon source amount used (and consequently the carbon/nitrogen ratio). In this regard the potential of this yeast as oleaginous was not completely explored.

In the next line a few comments and suggestions are made:

Line 73 – The introduction would gain if a better context of the strain used would be given and why explore it as a possible oleaginous yeast is here interesting and innovative.

Line 134 – I believe you meant that the suspension was filtered by passing through micro-fiber-glass filter and then dried (what temperature/time?)

Line 171 – Is there any possible and reasonable explanation on why the pseudomycelium shape is observed when glucose is used, but not on glycerol?

Line 103 – Is there any rational for the concentration of glucose and glycerol of 10 g/L?

Line 202 – In line with previous question, glucose is totally consumed in less than 24 hours (we do not know when, since there is no other intermediate point between 0 and 24 hours). Even so, although this lack of carbon source after 24 hours, the lipids accumulation continues until 72 hours, as well as biomass growth. It would be great that an explanation for these phenomena would be given in the discussion of these results.

The quality of English language is acceptable. 

Author Response

The authors greatly appreciate the time and effort invested by the reviewers in reading and considering the manuscript. The suggestions helped us to improve the writing and content of the article. We hereafter present a response to each observation.

Reviewer #1

This study reports the cultivation of Candida wangnamkhiaoensis (CW) to produce lipids from glucose and glycerol. The lipids are characterized regarding the fatty acid profiles obtained and compared with other traditional lipid sources. The methodology is well described and executed, and the results are clearly and carefully exposed. The manuscript is well structured and is easy to follow and understand.

Nevertheless there is no further exploration of factors that might contribute to improve the lipids production (which I assume was not the purpose of the study) – example, the increase of the carbon source amount used (and consequently the carbon/nitrogen ratio). In this regard the potential of this yeast as oleaginous was not completely explored.

Response: We appreciate this observation and agree with the reviewer that the oleaginous potential of this yeast was not completely explored. Indeed, among the diverse factors that could improve the lipid production and profile of fatty acids are the following:

  1. The composition of the culture medium, including the type and concentration of nutrients (the source of carbon, nitrogen, phosphorous, magnesium, zinc, calcium, vitamins, etc.), the C/N and C/P ratios, etc. In general, relatively high C/N and C/P ratios (providing a lower supply of nitrogen and phosphorous) favor the accumulation of lipids.
  2. Temperature
  3. pH
  4. Shaking
  5. Concentration of salts
  6. Growth phase and the age of the cultivation
  7. Level of dissolved oxygen

However, as the reviewer mentioned, the aim of the current study was not to explore the factors that could lead to the maximum production of lipids by CW. Research in this sense is presently underway and the results will be reported in due time.

Line 73 – The introduction would gain if a better context of the strain used would be given and why explore it as a possible oleaginous yeast is here interesting and innovative.

Response: Our group has studied the capacity of CW to produce α-amylase systemically in batches (Hernández-Montañez et al., 2012) and continuously (Chávez-Camarillo et al., 2018; Chávez-Camarillo et al., 2022), having reported some biochemical and molecular properties of the latter enzyme (Hernández-Montañez et al., 2012). During such research, we observed that CW is capable of accumulating lipids. To our knowledge, there are no reports to date on this capacity of the yeast. Hence, an evaluation of the potential of CW for generating lipids and the corresponding fatty acid profile should certainly be of interest for the scientific community. Although there is a recent report on CW, its metabolic and physiological capacities as well as its potential in biotechnological processes are unknown.

We have included this explanation in the new version of the manuscript (lines 78-85).

Line 134 – I believe you meant that the suspension was filtered by passing through micro-fiber-glass filter and then dried (what temperature/time?)

Response: The reviewer is absolutely right. The corresponding sentence was rewritten and now reads (lines 142-144):

The yeast suspension was filtered through micro-fiberglass filters (Cytiva Whatman, St. Louis, MO, USA). The resulting biomass was washed twice with type II water, oven-dried at 60 °C for 24 h, and then quantified.

Line 171 – Is there any possible and reasonable explanation on why the pseudomycelium shape is observed when glucose is used, but not on glycerol?

Response: As mentioned in the original manuscript, diverse environmental conditions can induce a dimorphic transition from the unicellular yeast-like form of yeasts to the multicellular pseudomycelium form. Among such conditions, particularly important are changes in temperature or pH, and the level of dissolved oxygen and/or the presence of specific compounds in the culture medium. The latter compounds include carbon, nitrogen, amino acids, vitamins, and metallic ions, in which both their type and concentration are relevant (Gancedo, 2001; Ruiz-Herrera and Sentandreu, 2002; Boyce and Andrianopoulos, 2015; Kokoreva et al., 2022). Additionally, the dimorphic transition is dependent on the yeast species. Due to its medical and industrial importance, dimorphic switching continues to be studied in order to fully understand the controlling molecular mechanisms (Boyce and Andrianopoulos, 2015; Gautier, 2017; Zheng et al., 2021).

Considering the current state of knowledge, it is unknown why CW presently took on a pseudomycelium growth in glucose medium and a unicellular growth in glycerol medium. Nevertheless, these results are in agreement with those reported for Yarrowia lipolytica, a yeast with biotechnological potential. The latter yeast also shows a predominantly mycelium form in a medium of glucose plus (NH4)2SO4 and a yeast form in a medium of glycerol plus (NH4)2SO4 (Ruiz-Herrera and Sentandreu, 2002).

The information about Yarrowia lipolytica is now included in the manuscript (lines 199-204).

Line 103 – Is there any rational for the concentration of glucose and glycerol of 10 g/L?

Response: In our previous studies in which it was observed that CW produced lipids, the concentration utilized for the sources of carbon (glucose, maltose, soluble starch, and glycerol) was 10 g/L. Hence, we decided to employ the same concentration of glucose and glycerol in the present study. Further research is needed to encounter the optimal conditions for the production of lipids and oleic acid, including the best concentration of glucose and glycerol. We are currently investigating these factors.

Line 202 – In line with previous question, glucose is totally consumed in less than 24 hours (we do not know when, since there is no other intermediate point between 0 and 24 hours). Even so, although this lack of carbon source after 24 hours, the lipids accumulation continues until 72 hours, as well as biomass growth. It would be great that an explanation for these phenomena would be given in the discussion of these results.

Response: This explanation is included in the new version of the manuscript (lines 228-233), and now reads:

CW showed exponential growth during the first 24 h of cultivation with glucose (Figure 3b) and a slower growth rate afterwards when the substrate was nearly exhausted in the culture medium. Thus, CW was capable of growing and accumulating lipids under conditions of glucose scarcity, which may be due to its capability of intracellular accumulation of a carbohydrate reserve that can be used later (Chávez-Camarillo et al., 2022). That is, the yeast was previously prepared as a preinoculate grown in YPG, during which time it could accumulate a reserve of carbohydrates. This is also a likely explanation for the lack of cell lysis after 24 h of growth in the glucose medium. The maximum concentration of biomass of CW cultivated in glucose was found at 72 h (3.35 g L-1).

References

Boyce, K.J.; Andrianopoulos, A. Fungal Dimorphism: The Switch from Hyphae to Yeast Is a Specialized Morphogenetic Adaptation Allowing Colonization of a Host. FEMS Microbiology Reviews 2015, 39, 797–811, doi:10.1093/femsre/fuv035.

Chávez-Camarillo, G.M.; Lopez-Nuñez, P.V.; Jiménez-Nava, R.A.; Aranda-García, E.; Cristiani-Urbina, E. Production of Extracellular α-Amylase by Single-Stage Steady-State Continuous Cultures of Candida wangnamkhiaoensis in an Airlift Bioreactor. PLOS ONE 2022, 17, e0264734, doi:10.1371/journal.pone.0264734.

Chávez-Camarillo, G.Ma.; Santiago-Flores, U.M.; Mena-Vivanco, A.; Morales-Barrera, L.; Cortés-Acosta, E.; Cristiani-Urbina, E. Transient Responses of Wickerhamia sp. Yeast Continuous Cultures to Qualitative Changes in Carbon Source Supply: Induction and Catabolite Repression of α-Amylase Synthesis. Ann Microbiol 2018, 68, 625–635, doi:10.1007/s13213-018-1369-4.

Gancedo, J.M. Control of Pseudohyphae Formation in Saccharomyces cerevisiae. FEMS Microbiol. Rev. 2001, 25, 107–123, doi:10.1111/j.1574-6976.2001.tb00573.x.

Gauthier, G.M. Fungal Dimorphism and Virulence: Molecular Mechanisms for Temperature Adaptation, Immune Evasion, and In Vivo Survival. Mediators Inflamm 2017, 2017, 8491383, doi:10.1155/2017/8491383.

Hernández-Montañez, Z.; Juárez-Montiel, M.; Velázquez-Ávila, M.; Cristiani-Urbina, E.; Hernández-Rodríguez, C.; Villa-Tanaca, L.; Chávez-Camarillo, G. Production and Characterization of Extracellular α-Amylase Produced by Wickerhamia sp. X-Fep. Appl. Biochem. Biotechnol. 2012, 167, 2117–2129, doi:10.1007/s12010-012-9736-2.

Kokoreva, A.S.; Isakova, E.P.; Tereshina, V.M.; Klein, O.I.; Gessler, N.N.; Deryabina, Y.I. The Effect of Different Substrates on the Morphological Features and Polyols Production of Endomyces magnusii Yeast during Long-Lasting Cultivation. Microorganisms 2022, 10, 1709, doi:10.3390/microorganisms10091709.

Ruiz-Herrera, J.; Sentandreu, R. Different Effectors of Dimorphism in Yarrowia lipolytica. Arch Microbiol 2002, 178, 477–483, doi:10.1007/s00203-002-0478-3.

Zheng, F.; Gao, W.; Wang, Y.; Chen, Q.; Zhang, Q.; Jiang, X.; Hou, B.; Zhang, Z. Map of Dimorphic Switching‑related Signaling Pathways in Sporothrix Schenckii Based on Its Transcriptome. Molecular Medicine Reports 2021, 24, 1–10, doi:10.3892/mmr.2021.12285

Reviewer 2 Report

To the best of the authors' knowledge, it is the first report on lipid biosynthesis by Candida wangnamkhiaoensis. In my humble opinion, the authors provided a very good-written manuscript with appropriately designed research. The results were considered and clearly described, and the discussion was adequately presented. Taking into account the title of the manuscript, the production of oleic acid is nothing unusual in the case of yeast, but this was also described by the authors in Table 4. This monounsaturated acid is the fatty acid that is produced in the largest amounts by many species of yeast.

Author Response

The authors greatly appreciate the time and effort invested by the reviewers in reading and considering the manuscript. The suggestions helped us to improve the writing and content of the article. We hereafter present a response to each observation.

Reviewer #2

To the best of the authors' knowledge, it is the first report on lipid biosynthesis by Candida wangnamkhiaoensis. In my humble opinion, the authors provided a very good-written manuscript with appropriately designed research. The results were considered and clearly described, and the discussion was adequately presented. Taking into account the title of the manuscript, the production of oleic acid is nothing unusual in the case of yeast, but this was also described by the authors in Table 4. This monounsaturated acid is the fatty acid that is produced in the largest amounts by many species of yeast.

Response: The authors greatly appreciate these comments by the reviewer.